# Molecular Mechanisms and Pathophysiology of Myocardial Disease: Insights from Pediatric Inflammatory Multisystem Syndrome (PIMS) Associated with SARS-CoV-2

**DOI:** 10.3390/ijms26083580

**Published:** 2025-04-10

**Authors:** María Teresa Viadero, María Jesús Caldeiro, Natalia Fernández-Suarez, Jesús Garde, María Jesús Cabero, Domingo González-Lamuño

**Affiliations:** 1Pediatric Cardiology, Division of Pediatrics, University Hospital “Marqués de Valdecilla”, University of Cantabria, Avda. Valdecilla s/n, 39008 Santander, Spain; mteresa.viadero@scsalud.es (M.T.V.); natalia.fernandezs@scsalud.es (N.F.-S.); jesus.garde@scsalud.es (J.G.); 2Division of Pediatrics, University Hospital “Marqués de Valdecilla”, Avda. Valdecilla s/n, 39008 Santander, Spain; mjesus.caldeiro@scsalud.es; 3Head of Pediatrics Division, University Hospital “Marqués de Valdecilla”, University of Cantabria, Avda. Valdecilla s/n, 39008 Santander, Spain; 4Consultant of Nefrology, Rare Diseases and Clinical Genetics Department, Division of Pediatrics University Hospital “Marqués de Valdecilla”, University of Cantabria, Avda. Valdecilla s/n, 39008 Santander, Spain

**Keywords:** pediatric inflammatory multisystem syndrome (PIMS), multisystem inflammatory syndrome in children (MIS-C), myocardial dysfunction, cytokine storm, endothelial dysfunction, SARS-CoV-2, Kawasaki disease, biomarkers, IVIG, corticosteroids

## Abstract

Multisystem inflammatory syndrome in children (MIS-C), also known as pediatric inflammatory multisystem syndrome (PIMS), presents significant challenges in pediatric cardiology, due to its complex molecular pathophysiology. In this retrospective analysis of 15 cases that were managed at a single tertiary care center, we investigated the molecular contributors to myocardial dysfunction, including cytokine storms, hyperinflammation markers, and hypercoagulable states. Transient myocardial involvement was identified in 46.6% of patients, with complete recovery achieved within 2–4 weeks following treatment. Ferritin, NT-ProBNP, and troponin levels were significantly elevated in patients with ventricular dysfunction compared to those without. The neutrophil-to-lymphocyte ratio (NLR), which was previously identified as a severity marker in acute COVID-19, was also significantly higher in patients with ventricular dysfunction, suggesting its potential as a prognostic indicator in MIS-C. Notably, no coronary artery aneurysms were detected in the cohort. These findings underscore the importance of early, standardized therapeutic interventions in mitigating severe outcomes, and they provide valuable insights into the molecular mechanisms driving myocardial dysfunction in MIS-C. Incorporating NLR and ferritin into the initial diagnostic workup may improve the early triage and identification of high-risk MIS-C patients.

## 1. Introduction

Pediatric inflammatory multisystem syndrome (PIMS), also known as multisystem inflammatory syndrome in children (MIS-C), is a severe post-infectious complication of SARS-CoV-2, characterized by systemic inflammation and multi-organ involvement [1]. The cardiovascular system is particularly vulnerable, and some reports have described children with severe ventricular dysfunction or cardiogenic shock [1,2,3,4,5,6]. In contrast to Kawasaki disease (KD), which frequently results in coronary artery aneurysms [7], MIS-C is primarily associated with reversible myocardial injury [3,6].

Several studies suggest that the global rollout of SARS-CoV-2 vaccines has significantly reduced both the incidence and severity of MIS-C and also that vaccinated children, especially adolescents, generally experience milder disease trajectories [8,9,10]. These observations underscore the importance of understanding the molecular underpinnings of MIS-C in order to refine therapeutic strategies and optimize patient outcomes. Previous studies have identified hyperinflammation, cytokine storms [11,12], endothelial damage [13], and coagulation abnormalities [14] as key drivers of multiorgan and myocardial damage in MIS-C.

Despite these advances, significant gaps remain in terms of distinguishing the pathophysiology of MIS-C from other inflammatory conditions, such as Kawasaki disease. Accordingly, this study aims to elucidate the molecular mechanisms driving MIS-C and evaluate the impact of early, standardized interventions within a controlled clinical setting. Likewise, we carried out a comparison between patients with MIS-C who were with and without myocardial dysfunction.

Myocardial involvement in MIS-C arises from a complex interplay of molecular mechanisms. The pathophysiology may be multifactorial, potentially including direct injury to cardiomyocytes from SARS-CoV-2 viral invasion and the impact of a dysregulated immune response, leading to microvascular dysfunction and endothelial injury [11]. Hyperinflammation is a hallmark of MIS-C [11]. Previous studies have consistently reported an elevation in inflammatory markers (including C-reactive protein (CRP), the erythrocyte sedimentation rate (ESR), ferritin, procalcitonin, and IL-6 [15,16]. Other findings include an elevated white blood cell count with neutrophilia and lymphopenia, elevated D-dimer and fibrinogen, and elevated myocardial injury markers such as troponin and brain natriuretic peptide (BNP) [15,16]. Significant differences have been observed between these findings in MIS-C compared to those in severe COVID-19 cases [3].

Recent studies have highlighted the relevance of inflammatory markers, such as the neutrophil-to-lymphocyte ratio (NLR) and serum ferritin in COVID-19 and multisystem inflammatory syndrome in children (MIS-C). NLR, which is derived from routine blood counts, has been shown to correlate with disease severity, respiratory failure, and mortality in adults with COVID-19 [17,18]. Similarly, hyperferritinemia reflects macrophage activation and systemic inflammation and is commonly observed in hyperinflammatory syndromes, including MIS-C [3]. These markers, due to their accessibility and prognostic utility, may help stratify risk and guide therapeutic decisions in pediatric patients. Their evaluation in MIS-C could offer valuable insights into immune dysregulation and myocardial stress.

Elevated levels of pro-inflammatory cytokines, such as interleukin-6 (IL-6), interleukin-1β (IL-1β), and tumor necrosis factor-alpha (TNF-α), among others, have been found to be elevated in the context of MIS-C [11], which could initiate a systemic inflammatory response that directly contributes to myocardial injury. This cytokine storm is further compounded by SARS-CoV-2-induced endothelial dysfunction. The virus binds to angiotensin-converting enzyme 2 (ACE2) receptors that are present in the heart [11], leading to microvascular inflammation, thrombosis, and endothelial damage. Additionally, hyperactivation of the adaptive immune system results in the production of autoantibodies [19], which further exacerbates tissue injury. A concurrent hypercoagulable state—as evidenced by elevated D-dimer and fibrinogen levels [15,16]—amplifies cardiovascular stress and contributes to transient myocardial dysfunction.

These interconnected processes not only explain the nature of myocardial dysfunction in MIS-C but also show differences from that described in Kawasaki disease [3,12,19]. The pathophysiology of coronary enlargement in MIS-C has not been elucidated. However, its mild severity and rapid resolution may suggest that coronary enlargement in MIS-C more often results from vasodilation in the setting of a highly proinflammatory milieu, rather than from destruction of the arterial wall by inflammatory cells [3].

In this study, we aim to elucidate these molecular mechanisms by analyzing the clinical presentation, laboratory findings, therapeutic approaches, and outcomes in a case series of 15 pediatric patients treated at a single tertiary care center. By comparing our findings with the existing literature, including studies on Kawasaki disease and those with larger cohorts, like the MUSIC study [5], we seek to highlight the unique pathophysiology and management outcomes of PIMS/MIS-C.

## 2. Results

### 2.1. Demographics

The cohort comprised 15 patients with a median age of 10 years (range: 12 months to 15 years). Gender distribution was nearly balanced, with 53.3% male patients and 46.6% female patients. The majority (74%) were previously healthy, while 26% had a history of asthma or atopy. Additionally, 13% were overweight, and 6.5% presented with obesity. The series was homogeneous in terms of racial distribution (93.3% were Caucasian/Iberian). To establish the socioeconomic level, we used the TSI (Individual Health Card). This is a document that identifies users of the National Health System (NHS) and determines their level of copayment for medications [20] and family contribution to the payment of medicines, as described in the Spanish national health system, which graduates the contribution from 1 to 6, graded from lowest to highest, according to the family’s income. The mean TSI number was 3; this grade indicates that the user is an insured person with an income below EUR 18,000 per year and who must, therefore, pay 40% of the cost of subsidized medications (Table 1 summarizes the main epidemiological, clinical, analytical characteristics, treatments, and outcomes of the cohort).

### 2.2. Clinical Presentation and Laboratory Data

All patients presented with persistent high fever; almost all exhibited abdominal pain, vomiting, diarrhea, and malaise. Of this cohort, 60% met the Kawasaki criteria, according to Kawasaki guidelines [7,21]. Other main signs and symptoms are described in Table 1. Symptoms of myocardial dysfunction included chest discomfort, palpitations, and hypotension. Myocardial dysfunction (LVEF < 55%) was present in 46.6% of patients. Six patients (40%) developed shock, requiring intensive care support. Among them, nine were admitted to the Pediatric Intensive Care Unit (PICU), and five required vasoactive support. Inflammatory markers were universally elevated, including CRP (median: 200 mg/L), ESR (median 67 mm/h) and IL-6 (elevated in all analyzed cases (all of them with ventricular dysfunction). Ferritin was elevated to 40%. Overall, 80% had leukocytosis (higher counts than those established by the laboratory for that age range), 100% had neutrophilia (range: 78–96%), 80% had lymphopenia (range 100–800), 33.3% had thrombopenia (<150,000/mm^3^) upon admission, and 86.6% had thrombocytosis (>450,000/mm^3^) in the subacute phase. The presence of thrombocytopenia was more frequent in the group with ventricular dysfunction (71.5% versus 12.5%), but without statistically significant counts. Regarding the other blood count values, we also found no significant differences. None met the criteria for hemophagocytic syndrome. Coagulopathy was evident in all patients, as indicated by elevated D-dimer and fibrinogen levels, without statistically significant differences being found between patients with or without ventricular dysfunction.

Transient myocardial dysfunction (LVEF < 55%) was observed in 7 of 15 patients (46.6%), with a reduced left ventricular ejection fraction (LVEF) (median: 43.5%; IQR: 36.5–45%) detected in all seven cases. NT-proBNP values were elevated in 100% of patients (median: 4500 pg/mL; IQR: 2500–6000 pg/mL) and troponin was elevated in 53.3% of patients (median: 756 ng/L; IQR: 400–1300 ng/L). Ferritin, NT-ProBNP, and troponin levels showed statistically significant differences between both groups (with or without ventricular dysfunction). All patients achieved complete recovery of myocardial function within 2–4 weeks, with no coronary artery aneurysms (z-score > 2.5 DS) [7] or residual cardiac dysfunction detected at follow-up. None presented serious arrhythmias, although 20% had pericardial effusion (mild and quickly resolved in all cases). Table 2 compares biomarker levels between patients with and without myocardial dysfunction.

### 2.3. Therapies and Interventions

Therapeutic approaches followed national and international guidelines [22,23,24,25]. All patients received IVIG and all but one (93.3%) were treated with corticosteroids as first-line therapy. The mean duration of illness before treatment initiation was 4.8 days (range: 2–7 days). Two cases (13.3%) were classified as refractory, being defined as persistent fever and/or significant end-organ involvement despite initial immunomodulatory treatment. Typically, MIS-C patients respond to IVIG and glucocorticoids within the first 24 h, and intensification therapy is recommended for those who fail to show clinical improvement [24]. In our study, persistent fever was defined as lasting beyond 36 h after completing the initial IVIG dose, in accordance with the classic Kawasaki disease criteria, as fever before this timeframe may reflect an adverse reaction to IVIG rather than true refractoriness [7,21].

Both refractory patients received a second IVIG dose combined with high-dose corticosteroids (30 mg/kg), which successfully controlled fever and inflammatory markers without complications. The absence of complications in our cohort is particularly relevant, as the 2022 American College of Rheumatology (ACR) guidelines advise against IVIG retreatment due to the increased risk of hemolytic anemia [24], which is reported to be more frequent in MIS-C than in Kawasaki disease, leading to its contraindication.

Additionally, aspirin was administered in 93.3% of cases, while 53.3% of patients received anticoagulation prophylactic therapy due to increased thrombotic risk. None developed thrombosis. Inotropic support was required for five patients (33.3%). Two patients (13.3%) required high-flow oxygen therapy.

### 2.4. Outcomes

Full recovery of myocardial function was observed in all patients before discharge, with no fatalities or residual cardiac dysfunction at follow-up. None exhibited a coronary aneurysm or long COVID-19, based on the CDC definition [26]. Table A1 (Appendix A) includes the main patients data. 

### 2.5. Neutrophil-to-Lymphocyte Ratio (NLR) as a Potential Prognostic Marker

Although our study primarily focused on classical inflammatory and myocardial stress biomarkers (e.g., CRP, IL-6, NT-proBNP, and troponin), it is worth noting that recent evidence supports the neutrophil-to-lymphocyte ratio (NLR) as a simple, yet powerful, predictor of adverse outcomes in COVID-19, including MIS-C. NLR is calculated by dividing the absolute neutrophil count by the absolute lymphocyte count, both of which are readily available in routine blood tests.

In our pediatric cohort, 100% of patients presented with neutrophilia and 80% with lymphopenia at admission, strongly suggesting elevated NLRs. While we did not initially analyze this biomarker, retrospective analysis showed that NLR values were significantly higher in patients with myocardial dysfunction at all time points: T0 (14.6 vs. 7.4; *p* = 0.001), T1 (4.15 vs. 2.02; *p* = 0.025), and T2 (7.8 vs. 3.9; *p* = 0.005).

To further illustrate these differences, Table 3 summarizes the values of NLR and absolute leukocyte counts at admission (T0), on the median hospitalization day (T1), and at discharge (T2) in patients with and without myocardial dysfunction. The NLR was significantly elevated at all time points in patients with ventricular dysfunction, while absolute counts of neutrophils and lymphocytes did not differ significantly between groups.

## 3. Discussion

This study underscores the critical importance of early, standardized interventions for achieving full recovery from MIS-C. Notably, our cohort exhibited no coronary artery complications, a finding that contrasts with the MUSIC study [5], where 4.4% of patients developed coronary aneurysms—including at least one giant aneurysm (Table 4). This discrepancy likely reflects the variability in clinical care available across the 32 centers in the MUSIC study, which may have led to delays in diagnosis and inconsistencies in therapeutic approaches. Similarly, a multicenter study by Acevedo et al. [27] in a lower-income country reported markedly higher rates of shock (87%), reduced left ventricular ejection fraction (35%), coronary aneurysms (35%), and mortality (9%). The authors attributed these outcomes to limited healthcare access, which likely delayed diagnosis and prompt intervention, thereby facilitating disease progression and resulting in more severe hemodynamic complications compared to our cohort.

MIS-C and Kawasaki disease appear to be distinct clinical entities, differing notably in both pathophysiology and cardiovascular involvement [3,12,19]. Kawasaki disease is characterized by sustained vasculitis that predisposes patients to coronary artery aneurysms and long-term vascular remodeling [7]. In contrast, MIS-C is driven by a cytokine-mediated hyperinflammatory response that results in transient myocardial dysfunction. Elevated NT-proBNP and troponin levels in MIS-C indicate acute myocardial stress, distinguishing it from the chronic vascular alterations seen in Kawasaki disease [6,28]. These observations suggest that Kawasaki disease involves prolonged coronary inflammation and endothelial activation, whereas MIS-C is marked by rapid, cytokine-driven immune dysregulation.

A key strength of our study is the uniform management of all studied patients at a single tertiary care center, which ensured consistent diagnostic criteria, early detection, and the prompt initiation of treatment protocols. Our cohort’s mean time to treatment initiation was 4.8 days (range: 2–7 days), slightly shorter than the 5.2 days reported in the MUSIC study [5], a factor that may have contributed to the favorable coronary outcomes observed. Moreover, nearly all patients received intravenous immunoglobulin (IVIG) and corticosteroids as first-line therapy, effectively mitigating the inflammatory cascade and preventing coronary complications. In contrast, the MUSIC study reported lower corticosteroid usage (78.3%) and the heterogeneous application of alternative anti-inflammatory agents such as anakinra or infliximab, which may have influenced disease progression [5].

These findings emphasize the protective role of centralized, standardized care in minimizing treatment delays and ensuring timely immunomodulation. Such an approach could be critical in reducing the risk of coronary complications and long-term cardiovascular sequelae in MIS-C, as has already been demonstrated in Kawasaki disease [7].

The contrast with the MUSIC study [5], where participants were treated at 32 hospitals across North America, underscores how regional and institutional differences in clinical practice, diagnostic timing, and management strategies can impact outcomes. Differences in patient demographics, such as age distribution, underlying comorbidities, and healthcare accessibility, may also contribute to variations in disease progression and treatment responses. For instance, patients treated in larger, more diverse healthcare systems may experience diagnostic delays or inconsistent therapeutic interventions, which could influence clinical outcomes.

Furthermore, cultural, and systemic factors, including healthcare policies, resource availability, and clinician familiarity with emerging MIS-C guidelines, likely shaped the differences observed between the two studies. It should also be noted that our cohort was much more homogeneous in terms of race, socioeconomic level, and access to the health system (almost all were Caucasian, the majority from low- or middle-income families but none were in poverty, and, universally, were users of the NHS) than the MUSIC cohort, which could enable better results. Our single-center design allowed for streamlined treatment protocols and reduced variability, which may have minimized the risk of coronary complications. This model of centralized, uniform care could serve as a benchmark for other healthcare systems, demonstrating how standardization can mitigate variability and improve patient outcomes. Conversely, the multicenter design of the MUSIC study introduced heterogeneity, reflecting real-world challenges in terms of standardizing care across diverse settings. The MUSIC study further supports the reversibility of myocardial dysfunction in MIS-C, reporting that 99% of participants normalized their left ventricular systolic function within six months. This aligns with our findings, as all the patients in our cohort achieved full recovery.

The clinical presentation of MIS-C has often been compared to Kawasaki disease due to certain overlapping features, including fever, mucocutaneous involvement, and cardiac complications [25]. However, our literature review suggests that only one-quarter to one-half of MIS-C patients meet the full Kawasaki disease diagnostic criteria [22]. In our cohort, 33% (5/15) of patients met the classic (complete) Kawasaki disease criteria according to the American Heart Association (AHA) definition [23].

Comparisons with Kawasaki disease highlight distinct underlying immune mechanisms. While Kawasaki disease is characterized by prolonged coronary inflammation, MIS-C represents a transient, cytokine-driven hyperinflammatory state. The elevated levels of NT-proBNP and troponin observed in patients with myocardial dysfunction in our cohort indicate acute myocardial stress and injury, which are hallmark features of MIS-C. These biomarkers serve as critical tools for the early recognition of cardiac involvement and reflect the transient, cytokine-driven myocardial depression that characterizes this syndrome [4,29]. In contrast to Kawasaki disease, where myocardial damage is often linked to persistent coronary inflammation and vascular remodeling, MIS-C typically presents with the rapid onset of dysfunction and recovery following immunomodulatory therapy [30].

Ferritin levels were notably elevated among patients with myocardial involvement, reinforcing its role as an inflammatory marker in MIS-C. As an acute-phase reactant and iron-storage protein, ferritin has gained recognition as a valuable biomarker in MIS-C, particularly due to its association with macrophage activation and hyperinflammation [3]. In our cohort, patients with myocardial dysfunction demonstrated significantly higher ferritin levels compared to those without cardiac involvement. This finding suggests that ferritin may serve not only as an early indicator of cardiac involvement but also as a potential marker to guide therapeutic escalation. The elevated ferritin levels observed in MIS-C likely reflect a macrophage-driven inflammatory response, a mechanism implicated in the pathophysiology of hemophagocytic lymphohistiocytosis (HLH), a condition with clinical features overlapping those of MIS-C. Although none of our patients met the diagnostic criteria for HLH, the pronounced ferritin elevation recorded, particularly in those with cardiac dysfunction, underscores its potential utility for identifying high-risk cases. Elevated ferritin levels have been associated with macrophage activation syndrome and poorer outcomes in both severe COVID-19 and MIS-C [31,32], further supporting its prognostic relevance in this setting.

Although the difference in platelet counts between groups was not statistically significant, a trend toward thrombocytopenia was observed in the myocardial dysfunction group, aligning with the findings from other MIS-C studies describing platelet consumption in inflammatory states [33]. These biomarker profiles underscore the immunopathogenic differences between MIS-C and Kawasaki disease. While KD involves chronic endothelial activation and a significant risk of coronary aneurysms, MIS-C is characterized by acute elevations in proinflammatory cytokines—such as IL-6, IL-1β, and TNF-α—leading to systemic inflammation and transient cardiac dysfunction [28,34]. The absence of coronary aneurysms in our cohort despite evident myocardial involvement supports the hypothesis of a rapid, cytokine-mediated inflammatory response rather than sustained vascular injury. Shock in Kawasaki disease is rare, at less than 5–10%, and is associated with a greater risk of coronary artery abnormalities and resistance to immunoglobulin therapy [35]. These distinctions underscore different immune pathways: Kawasaki disease is primarily linked to sustained inflammation and endothelial activation, whereas MIS-C reflects rapid, cytokine-driven immune dysregulation [6,28].

The consequent prolonged activation of the innate immune system and the continuous suppression of lymphocytes released into the circulation are the main characteristics of COVID-19 disease. An elevated absolute neutrophil count has been described as a negative predictor of outcome in COVID-19 patients, reflecting the hyper-activation of the innate immune response caused by both virus-triggered or cytokine-dependent mechanisms. In particular, neutrophils are involved in platelet activation, the over-production of inflammatory cytokines, and epithelial and endothelial cell damage [18]. In contrast, the adaptative immune response induces a reduction in absolute lymphocyte count in COVID-19 as a consequence of extended TNF-α-induced apoptosis, peripheral consumption, a direct ACE-2-cytopathic effect, or through interaction with CD147. Peripheral lymphopenia in COVID-19 patients has been largely described in the recent literature, which assessed its predictive value for disease severity and mortality. In addition, neutrophilia itself leads to the suppression of lymphocytes through a cytotoxic indirect effect [18]. NLR represents the balance between innate and immune responses. It readily increases as a consequence of a physiological and pathophysiological response to acute stress. NLR may be considered a marker of subclinical inflammation [18]. Although our study primarily focused on classical inflammatory and myocardial stress biomarkers (e.g., CRP, IL-6, NT-proBNP, and troponin), it is worth noting that recent evidence supports the neutrophil-to-lymphocyte ratio (NLR) as a simple, yet powerful, predictor of adverse outcomes in COVID-19, including MIS-C. In adult populations, the authors of [17,18] demonstrated that NLR is independently associated with ICU admission, respiratory failure (as reflected by the PaO_2_/FiO_2_ ratio), and mortality, outperforming C-reactive protein (CRP) in some cases. Specifically, while CRP was more predictive of PICU admission, NLR showed a stronger correlation with mortality and an inverse relationship with oxygenation indices (PaO_2_/FiO_2_), indicating its role as an early marker of immune dysregulation and poor prognosis.

In our pediatric cohort, 100% of patients presented with neutrophilia and 80% with lymphopenia at admission, strongly suggesting elevated NLRs. While we did not initially analyze this biomarker, retrospective analysis showed that NLR values were significantly higher in patients with myocardial dysfunction at all time points (admission, mean stay, and discharge). These differences suggest that NLR could serve as a sensitive marker of myocardial involvement in MIS-C. Building on the recent literature, we further propose that the neutrophil-to-lymphocyte ratio (NLR) may serve as a promising prognostic marker in MIS-C. Our retrospective data analysis demonstrated significantly higher NLR values at all time points among patients with myocardial dysfunction, mirroring the trends described in adult COVID-19 patient populations [17,18]. These findings support the utility of NLR as a readily available and dynamic indicator of immune dysregulation, particularly when traditional markers may be delayed or unavailable.

Incorporating NLR and ferritin into the routine initial workup for MIS-C could enhance the early triage and identification of high-risk patients. Their availability, simplicity, and low cost make them valuable additions to the biomarker panel already used to guide therapy. Future prospective studies with larger cohorts should validate the predictive power of these markers and establish threshold values that correlate with outcomes such as PICU admission, myocardial dysfunction, or the need for vasoactive support.

Acknowledgment of Limitations. This study is limited by its small sample size of 15 patients, which impacts the generalizability of the findings and introduces potential sampling bias. The single-center design may further lead to institutional biases, limiting the broader applicability of the results across diverse, multicenter populations. Additionally, the lack of long-term follow-up data restricts our ability to draw conclusions about potential delayed cardiovascular complications, such as residual myocardial dysfunction or coronary abnormalities. Comparisons made with the MUSIC cohort, while contextual, may be influenced by differences in diagnostic protocols and patient demographics, further highlighting the need for caution when extrapolating these results to compare larger populations.

## 4. Methods

### 4.1. Study Design and Setting

This retrospective case series was conducted at University Hospital Marqués de Valdecilla, HUMV, a single tertiary care hospital. It is a referral regional center that manages all patients diagnosed with MIS-C in our region. This study was conducted in accordance with the ethical principles for medical research involving human subjects set forth in the Declaration of Helsinki and was approved by the Ethics Committee for research concerning medicines and health products in Cantabria (CEIm Cantabria, code 2024.114). Written informed consent and assent were obtained from the patients’ parents or legal guardians. Assent from older pediatric patients was also obtained. Those who were lost to follow-up could be enrolled under a waiver consent.

All patients diagnosed with MIS-C in our region between April 2020 and August 2024 were included in the study. None declined to participate. Two patients were followed up until hospital discharge; further follow-up was lost because they were living in another Spanish region.

Clinical, laboratory, and imaging data were retrospectively collected from medical records. Diagnoses were based on the 2020 World Health Organization (WHO) [36] and updated 2023 Centers for Disease Control and Prevention (CDC) [37] criteria, which emphasize: (1) persistent fever lasting at least 24 h; (2) systemic inflammation, confirmed by elevated markers (CRP—C-reactive protein, ESR—erythrocyte sedimentation rate, or ferritin); (3) multi-organ involvement; (4) evidence of SARS-CoV-2 infection or exposure; and (5) no other plausible diagnosis.

The study cohort comprised 15 patients who were managed uniformly under standardized protocols [22,23,24,25], ensuring consistent therapeutic approaches. This adherence to WHO and CDC criteria [36,37] facilitated the direct comparison of our outcomes with those reported in international studies.

### 4.2. Data Collection

Clinical data included demographics, comorbidities, symptoms, and clinical and laboratory findings. Race and ethnicity data were collected. To assess the socioeconomic level, we used the TSI number for “contribution to the payment of medications in the NHS”. This is a number from 1 to 6 (lowest to highest), assigned according to family income: 1: minimum income, being unemployed without benefits; 2: some pensioners; 3: an income of <18,000 EUR/year; 4: an income of between 18,000 and 100,000 EUR/year; 5: an income of >100,000 EUR/year [20]. Data collected included the body mass index (BMI) and z-score, along with weight categories (normal/underweight—BMI z-score ≥ 1, overweight—z-score of >1 and ≤2, or obesity—z-score of >2) [38].

Laboratory findings focused on inflammatory (CRP, ESR, IL-6, and ferritin), cardiac biomarkers (NT-proBNP, and troponin), blood counts (white cells, red cells, and platelets), and coagulation profiles (D-dimer and fibrinogen). In addition to the standard laboratory parameters, we retrospectively calculated the neutrophil-to-lymphocyte ratio (NLR) at three time points: admission (T0), median hospitalization day (T1), and at discharge (T2)—defined as transfer to the Pediatric Intensive Care Unit, PICU, or routine discharge, using the formula NLR = neutrophils/lymphocytes.

The echocardiographic evaluation assessed left ventricular ejection fraction (LVEF) and coronary artery involvement. The initial evaluation was made upon the suspicion of MIS-C, and follow-up assessments were performed according to clinical evolution by guideline recommendations [22,23,24,25].

Echocardiograms were performed interchangeably by three experienced pediatric cardiologists in the Marqués de Valdecilla University Hospital (HUMV), following the American Society of Echocardiography (ASE) guidelines [39]. Standardized assessments of left ventricle (LV) diameters and volumes in diastole and systole were made from parasternal and apical imaging, and LV ejection fraction (LVEF) was calculated according to these values. Considering it is a pediatric cohort and similar to the MUSIC study [5], an LVEF of ≥55% was considered normal, an LVEF of 45–54% was considered mildly reduced, an LVEF of 35–44% was considered moderately reduced, and an LVEF of <35% was severely reduced. The dimensions of the left main coronary artery (LMCA), proximal left anterior descending artery (LAD), proximal circumflex artery (CA), and right coronary artery (RCA) were measured from the parasternal long and short axis and subcostal imaging. Coronary artery aneurysms were categorized as ectasia (z-score ≥2 to <2.5), small aneurysm (z-score 2.5 to <5), medium aneurysm (z-score 5 to ≤10), and giant aneurysm (z-score > 10 and/or a diameter of >8 mm) [7]. Z-scores were calculated using the Boston [40] and Kobayashi equations [41]. It was not necessary to perform computerized tomography (CT) or angiography for coronary assessment because the echocardiographic images were adequate in all cases.

### 4.3. Interventions

All patients, as detailed in Table 1, received standardized treatment according to national and international guidelines [22,23,24,25], including intravenous immunoglobulin (IVIG) and corticosteroids as first-line therapies. Anticoagulation and aspirin were administered, based on thrombotic risk [25]. Patients with shock received inotropic support and fluid resuscitation [25]. Follow-ups at six months ensured the resolution of cardiac dysfunction.

### 4.4. Outcome Measures

The primary outcome was myocardial recovery, which can be defined as the normalization of LVEF within 2–4 weeks of treatment. Secondary outcomes included the absence of coronary artery aneurysms and the clinical resolution of symptoms. At follow-up, no controlled patients showed subsequent symptoms or ventricular dysfunction.

### 4.5. Statistical Analysis

Data were summarized using descriptive statistics. Continuous variables such as biomarker levels and LVEF were reported as means, medians, and interquartile ranges (IQR), while categorical variables were expressed as frequencies and percentages. Statistical analyses included the chi-square tests, Fisher’s exact test for categorical variables, and the Mann–Whitney U tests for continuous variables. Statistical comparisons were performed between patients with and without myocardial dysfunction using Mann–Whitney U tests, considering *p*-values of <0.05 as statistically significant.

## 5. Conclusions

In our cohort, which was uniformly managed in accordance with international guidelines, nearly half of the patients presented with ventricular dysfunction, which resolved completely without mortality or coronary artery complications. Inflammatory, myocardial injury, and coagulation markers were universally elevated, with ferritin, NT-proBNP, and troponin levels being significantly higher among those patients with ventricular dysfunction. Retrospective analysis also revealed a significantly elevated neutrophil-to-lymphocyte ratio (NLR) in this subgroup, suggesting its potential role as a prognostic marker of cardiac involvement.

These findings underscore the effectiveness of early, standardized therapeutic approaches—particularly intravenous immunoglobulin and corticosteroids—in achieving full cardiac recovery and preventing long-term complications in MIS-C. The absence of coronary artery aneurysms further supports the benefit of prompt intervention, although the study’s limited sample size warrants cautious interpretation. Future research should explore biomarker-guided treatment algorithms, long-term cardiac outcomes, and the role of genetic predisposition and cytokine-targeted therapies in refining disease management. This study also highlights NLR and serum ferritin as accessible, cost-effective tools for the early identification of myocardial dysfunction in MIS-C, especially in resource-limited settings. Prospective studies are needed to validate their prognostic utility and inform personalized therapeutic strategies.

## Figures and Tables

**Table 1 ijms-26-03580-t001:** Cohort characteristics and outcomes.

Feature	Observation
Total Patients	15
Gender Distribution (%)	
Males	53.3%
Females	46.6%
Race/Ethnicity (%)	
Caucasian/Iberian	93.3%
Hispanic/Latino	6.6%
Socioeconomic level—TSI *—(% of patients)	
1	6.6%
2	0%
3	60%
4	20%
5	6.6%
6	6.6%
Median Age (mean)	10 years (range: 12 months–15 years)
Microbiological Positive PCR SARS-CoV-2 Positive SARS-CoV-2 serologies: -IgM -IgG -Both	33.3% 100% 0% 80% 20%
Common Symptoms (%): Fever Malaise Abdominal pain Vomiting Diarrhea Oral or pharyngeal changes Exanthema Kawasaki criteria ** Headache Fingers peeling Conjunctivitis	100% 93% 80% 80% 73% 66% 60% 60% (complete: 33.3%, incomplete 26.6%) 46% 40% 33%
Myocardial Involvement (%)	46.6% (7/15 patients)
Biomarkers (%)	
Elevated CRP	100%
IL-6 (*n* = 4)	100% of those analyzed
Ferritin	40%
NT-proBNP	100%
Troponin	53.3%
Fibrinogen	93.3%
D-dimer	100%
Treatments (%)	
IVIG	100%
Corticosteroids	93.3%
Aspirin	93.3%
Anticoagulants	53.3%
Outcomes (%)	
Full recovery	100%
Coronary aneurysms	0%
Persistent/Long COVID-19	0%

Legend: This table summarizes the demographic clinical features, biomarkers, received treatments, and outcomes of the 15 patients in the cohort. * TSI (contribution to the payment of medications in the NHS, coded from 1 to 6, indicating lowest to highest according to family income. 1: minimum income, unemployed without benefits; 2: some pensioners; 3: income < 18,000 EUR/year; 4: income between 18,000 and 100,000 EUR/year; 5: income >100,000 EUR/year). ** Kawasaki disease criteria, according to the AHA guides. CRP (C-reactive protein), ESR (erythrocyte sedimentation rate), and IL-6 (interleukin-6) are markers of inflammation; NT-proBNP (N-terminal pro-brain natriuretic peptide) and troponin indicate myocardial stress, while fibrinogen and D-dimer indicate coagulopathy. Key findings include a balanced gender distribution, a median age of 10 years, and a 46.6% rate of myocardial involvement. Notably, all patients achieved full recovery without coronary aneurysms. IVIG = Intravenous immunoglobulin.

**Table 2 ijms-26-03580-t002:** Biomarker level comparisons between patients with and without myocardial dysfunction.

Biomarker	Patients with Myocardial Dysfunction	Patients Without Myocardial Dysfunction	Significance
PCR (mg/dL)	15.9 (6.2–21.4)	14.7 (5.1–29.2)	*p* > 0.05
ESR (mm/h)	66.1 (15–120)	68.6 (17–101)	*p* > 0.05
Ferritin (mg/dL)	355 (59–733)	160 (79–362)	* *p* = 0.038
NT-ProBNP (pg/mL)	7223 (671–14,800)	1906 (618–4213)	* *p* = 0.001
Troponin (ng/mL)	1504 (101–6222)	12 (2–64)	* *p* < 0.001
D-dimer (ng/mL)	3240 (1400–6017)	3855 (1708–12,827)	*p* > 0.05
Fibrinogen (mg/mL)	789 (400–1149)	801 (684–1265)	*p* > 0.05

Legend: This table compares biomarker levels (CRP—C-reactive protein, ESR—erythrocyte sedimentation rate, ferritin, NT-ProBNP—N-terminal pro-brain natriuretic peptide, troponin, and D-dimer) in our cohort between patients with and without ventricular dysfunction (mean value and IQR). The *p*-values were calculated using the Mann–Whitney U test, with a *p*-value of <0.05 indicating statistical significance, highlighted with a *.

**Table 3 ijms-26-03580-t003:** Comparison of the neutrophil-to-lymphocyte ratio (NLR) and leukocyte counts between patients with and without myocardial dysfunction.

Biomarker	Patients with Myocardial Dysfunction (*n* = 7)	Patients Without Myocardial Dysfunction (*n* = 8)	Significance
Lymphocyte (cells/μL)	571 (400–800)	2325 (100–11,000)	*p* = 0.55
Neutrophil (cells/μL)	14,102 (6950–22,500)	16,380 (7950–23,124)	*p* = 0.48
NLR T0	14.61 (4.7–43.0)	7.39 (2.4–16.7)	* *p* = 0.001
NLR T1	4.15 (1.54–14.8)	2.02 (0.5–5.04)	* *p* = 0.025
NLR T2	7.80 (2.58–12.8)	3.90 (0.8–6.8)	* *p* = 0.005

Legend: NLR = neutrophil-to-lymphocyte ratio. T0 = admission, T1 = median hospitalization day, and T2 = at discharge (transfer to PICU or discharge). Data are presented as the median (range). The *p*-values were calculated using the Mann–Whitney U test, with a *p*-value of <0.05 indicating statistical significance, highlighted with a *.

**Table 4 ijms-26-03580-t004:** Comparison with the MUSIC study (JAMA 2025).

Feature	Current Study	MUSIC Study (JAMA 2025)
Total Patients	15	1204
Coronary Artery Involvement	None	15 cases (1 giant aneurysm)
Myocardial Dysfunction	46.6% transient dysfunction	42% transient dysfunction
Recovery of LV Function	100%	99%
Treatments	IVIG, corticosteroids, aspirin, and heparin	IVIG and corticosteroids
Mortality	None	0.3%
Biomarkers	Elevated CRP, IL-6, NT-proBNP, and troponin	Elevated CRP, IL-6, NT-proBNP, and troponin

Legend: This table compares the findings of the current study with those from the MUSIC study, highlighting key differences in coronary artery involvement, recovery rates, and biomarkers such as CRP (C-reactive protein), IL-6 (interleukin-6), NT-proBNP (N-terminal pro-brain natriuretic peptide), and troponin. Our study’s centralized approach ensured uniform disease management, contributing to the absence of coronary complications, whereas the MUSIC study, conducted across 32 North American hospitals, reflected variability in treatment practices. IVIG = intravenous immunoglobulin.

## Data Availability

Clinical and analytical data have been obtained from digitized clinical records.

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
