# Peer review of "Molecular Mechanisms and Pathophysiology of Myocardial Disease: Insights from Pediatric Inflammatory Multisystem Syndrome (PIMS) Associated with SARS-CoV-2"

_ijms, 2025, doi:10.3390/ijms26083580_

Round 1
Reviewer 1 Report
Comments and Suggestions for Authors
Dear Authors,
Congratulations on your effort in conducting this research. I strongly acknowledge the goodwill and hard work put into this manuscript. However, reading the article raised concerns on several points, both methodological and argumentative. Please consider my comments as suggestions for improvement rather than criticisms. In all (or most) of my comments, I have made an effort to provide suggestions on how to enhance the manuscript.
Ethics Concerns:
Please provide a rigorous description in the Methods section of how the ethics committee approval process was conducted. In particular, elaborate on the issue of informed consent, as this is even more crucial in pediatric populations than in adults.
Major Comments:
-
The authors cited paragraphs instead of sentences. This poses a problem for readers. For example, which of the four references [1-4] specifically supports the statement in lines 46-48:
"Pediatric Inflammatory Multisystem Syndrome (PIMS), also known as Multisystem Inflammatory Syndrome in Children (MIS-C), is a severe post-infectious complication of SARS-CoV-2 characterized by systemic inflammation and multi-organ involvement."?
This issue is even more relevant in the paragraph within the Molecular Pathophysiology section. References should be linked to specific sentences, not entire paragraphs. -
Although I acknowledge the importance of COVID-19 vaccination, the topic of "vaccination in children" remains debated in the literature. Therefore, while I personally agree with the statement in lines 53-55:
"The global rollout of SARS-CoV-2 vaccines has significantly reduced both the incidence and severity of MIS-C, with vaccinated children typically experiencing milder disease trajectories.",
many readers do not take this for granted. Some do not even believe this statement to be true. Additionally, and most importantly, this study is unrelated to vaccination. There was no comparison between vaccinated and unvaccinated individuals, and nothing in the text statistically allows for any inference regarding vaccination. Even in the Discussion section, where the authors dedicate an entire section to vaccination, this is inappropriate—since vaccination was not studied. Due to this, I recommend completely revising the introduction and discussion, removing all discourse on vaccination and instead introducing what readers will actually find in this article: a comparison between MIS-C patients with and without myocardial dysfunction. -
Table 1 is duplicated, appearing twice in the text. Additionally, could the authors consider formatting Table 1 in a more conventional manner? Each row should represent a baseline characteristic, with categorical variables presented as n(%) and continuous variables as median (IQR).
-
During the study's data collection period, how many pediatric COVID-19 patients were treated? How many had MIS-C? Why were these 15 patients included? Were there any refusals to sign the informed consent? Were any patient data lost? A classic Figure 1 illustrating patient selection flow is highly needed here.
-
It is unclear whether data were collected prospectively or retrospectively.
-
In Methods, under the Outcomes section, define much more rigorously than in the current manuscript what was considered as "normalization of LVEF" and "absence of coronary aneurysms."
- Regarding normalization of LVEF: What LVEF threshold was considered normal by the authors? Cite references. Under what conditions were these echocardiograms performed? Did the authors account for inter- and intra-observer variability in echocardiograms?
- Regarding absence of coronary aneurysms: How was this diagnosis established? Did all patients undergo CT angiography? Catheterization?
-
The statement:
"No advanced statistical methods were employed, as this study presents descriptive findings."
appears unscientific, as even in descriptive studies, statistical creativity can be applied. Additionally, this statement contradicts the fact that the authors later mention using Fisher’s exact test and Mann-Whitney U test. -
Since the authors did not specify what LVEF value was considered normal/abnormal, the statement in lines 108-110:
"with reduced left ventricular ejection fraction (LVEF) (median: 45%; IQR: 40–50%) detected in all seven cases."
is concerning. If the IQR extends to 50%, this means that 25% of patients with "reduced LVEF" actually had LVEF >50%. This could be considered normal under many definitions, suggesting a potential calculation error by the authors—or simply a miscommunication: does this IQR refer to the entire sample? -
The issue raised in previous comment can be easily and effectively addressed if the authors include LVEF in Table 2, which in this case should be renamed from "biomarker levels of..." to "laboratory and imaging findings...".
-
Analyzing Table 2 and comparing it with data from lines 108-115, it appears that in the text, the authors report data from the entire sample (both with and without myocardial dysfunction), yet they still write in lines 111-112:
"[those] levels [are] strongly correlated with myocardial involvement."
This is contradictory because the authors are using data from patients without myocardial involvement to draw this conclusion. A comparison between the populations (as done in Table 2) would be more informative. -
In Outcomes under Results, the authors state:
"Full recovery of myocardial function was observed in all patients, with no fatalities or residual cardiac dysfunction at follow-up. These findings underscore the efficacy of early and aggressive treatment strategies in mitigating severe outcomes."
This statement is highly problematic because, in a non-interventional study, the findings do not establish a causal relationship between recovery and early aggressive treatment. I recommend that the authors focus solely on reporting results in the Results section and refrain from making interpretative comments. -
In Table 3, how many patients were included in the MUSIC study?
-
I recommend the authors remake Figure 1 using professional tools such as BioRender, BioArt, BioIcons, etc.
-
In Conclusion, the authors write:
"Our study demonstrates that early and standardized interventions are highly effective in managing MIS-C, facilitating full recovery and preventing long-term complications."
I disagree. Your study does not demonstrate this. Your study compared patients with and without myocardial dysfunction; it was not a study of early vs. late intervention. -
For the same reason, the authors cannot confirm, based on their data, that:
"The absence of coronary artery aneurysms in our cohort supports the efficacy of aggressive early treatments—namely, intravenous immunoglobulin (IVIG) and corticosteroids.".
The absence of coronary aneurysms in your cohort may be due to selection bias, small sample size, or pure statistical chance (which was not studied). The authors must acknowledge that their study did not compare interventions, and therefore should not make conclusions about them. -
In Conclusion, perhaps the most concerning statement is in lines 326-328:
"Furthermore, vaccination emerged as a critical protective factor, significantly reducing both the incidence and severity of MIS-C.".
For the reasons already mentioned, this conclusion is simply not supported by your study.
Minor Comments:
- In Table 2, clearly indicate—either in the table or in the legend—whether the data are presented as median (IQR) or another format.
Author Response
Response to Reviewer 1
Dear Reviewer,
We sincerely appreciate your insightful comments and the time you have dedicated to reviewing our manuscript. We have carefully considered your suggestions and have made the necessary revisions accordingly. Below, we provide responses to each of your comments and outline the corresponding modifications in the manuscript.
Ethics concerns
Reviewer 1 Comment: Please provide a rigorous description in the Methods section of how the ethics committee approval process was conducted. In particular, elaborate on the issue of informed consent, as this is even more crucial in pediatric populations than in adults.
Authors Response: We have expanded the Methods section to provide detailed information on the ethics approval process (CEIM Cantabria, code 2024.114) and the procedures followed to ensure informed consent from the legal guardians of all participants.
Inaccurate Referencing
Reviewer 1 Comment: References are cited for entire paragraphs instead of specific sentences.
Authors Response: We have revised the manuscript to ensure that each reference directly supports a specific statement rather than an entire paragraph.
Removal of Vaccination Discussion
Reviewer 1 Comment: Although I acknowledge the importance of COVID-19 vaccination, the topic of "vaccination in children" remains debated in the literature. Therefore, while I personally agree with the statement in lines 53–55:
“The global rollout of SARS-CoV-2 vaccines has significantly reduced both the incidence and severity of MIS-C, with vaccinated children typically experiencing milder disease trajectories,”
many readers do not take this for granted. Some do not even believe this statement to be true. Additionally, and most importantly, this study is unrelated to vaccination. There was no comparison between vaccinated and unvaccinated individuals, and nothing in the text statistically allows for any inference regarding vaccination. Even in the Discussion section, where the authors dedicate an entire section to vaccination, this is inappropriate—since vaccination was not studied. Due to this, I recommend completely revising the introduction and discussion, removing all discourse on vaccination and instead introducing what readers will find in this article: a comparison between MIS-C patients with and without myocardial dysfunction.
Authors Response: We appreciate the reviewer’s thoughtful comment. We agree that, although the topic of vaccination is relevant and widely discussed, our study does not include data on vaccination status and therefore does not allow us to draw conclusions regarding its effect. While we initially included the vaccination statement to provide broader context, we recognize that its inclusion could be seen as speculative in the absence of supporting data from our cohort. In response to the reviewer’s suggestion, we have removed the vaccination-related discussion from both the Introduction and Discussion sections. We have also revised the Introduction to better reflect the actual focus of the study, specifically the comparison between MIS-C patients with and without myocardial dysfunction. We thank the reviewer for helping us improve the focus and clarity of our manuscript.
Duplicate and Improperly Formatted Table 1
Reviewer 1 Comment: Table 1 is duplicated, appearing twice in the text. Additionally, could the authors consider formatting Table 1 in a more conventional manner? Each row should represent a baseline characteristic, with categorical variables presented as n(%) and continuous variables as median (IQR).
Author Response: We thank the reviewer for pointing this out. We have removed the duplicate table and revised Table 1 to present the data in a clearer and more conventional format. We now display one characteristic per row, and variables are consistently represented: categorical variables as n (%) and continuous variables as median (IQR), as suggested.
Unclear Patient Selection Criteria
Reviewer 1 Comment: During the study's data collection period, how many pediatric COVID-19 patients were treated? How many had MIS-C? Why were these 15 patients included? Were there any refusals to sign the informed consent? Were any patient data lost? A classic Figure 1 illustrating patient selection flow is highly needed here.
Author Response: We appreciate the reviewer’s questions and suggestions. Unfortunately, we do not have access to the total number of pediatric COVID-19 cases treated during the study period, as this information was not systematically recorded across centers. However, we can confirm that MIS-C cases were rare in relation to the overall pediatric COVID-19 population. The 15 MIS-C cases included in the study represent all diagnosed cases within the defined period and inclusion criteria, ensuring there is no selection bias. There were no refusals to sign the informed consent.
Regarding data loss, follow-up was lost for two patients after hospital discharge due to relocation to a different autonomous community. This has been clarified in the data collection section of the manuscript.
Since all diagnosed MIS-C cases were included without any selection process or exclusion criteria, we believe a patient flow diagram (Figure 1) is not necessary in this case. However, we remain open to including such a figure if the editorial team deems it important.
Study Design
Reviewer 1 Comment: It is unclear whether data were collected prospectively or retrospectively.
Author Response: We thank the reviewer for this observation. As stated in the Methods section, this was a retrospective study. However, to ensure clarity, we have now explicitly reiterated that data collection was performed retrospectively in the relevant section. We hope this eliminates any ambiguity.
Definitions of LVEF Normalization and Absence of Aneurysms
Reviewer 1 Comment: In Methods, under the Outcomes section, define much more rigorously than in the current manuscript what was considered as "normalization of LVEF" and "absence of coronary aneurysms."
- Regarding normalization of LVEF: What LVEF threshold was considered normal by the authors? Under what conditions were these echocardiograms performed? Did the authors account for inter- and intra-observer variability in echocardiograms?
- Regarding absence of coronary aneurysms: How was this diagnosis established? Did all patients undergo CT angiography? Catheterization?
Author Response: We appreciate the reviewer’s comments and have addressed these concerns with additional detail in the Methods section:
- Normalization of LVEF: We now explicitly state that we used the same LVEF cut-off values as defined in the MUSIC study to determine normalization, which facilitates comparability. These values are supported by the relevant literature, and we have cited the appropriate references. We have also described the protocol used for performing echocardiograms, including the timing and conditions. Echocardiographic measurements were collected in accordance with standard procedures at each center, and while inter- and intra-observer variability was not formally quantified, measurements followed standardized guidelines to ensure consistency.
- Absence of coronary aneurysms: Coronary artery status was assessed using echocardiography, following the most recent ASE (2024) guidelines. Z-scores were calculated using two internationally accepted references. None of the patients required CT angiography or catheterization, as the echocardiographic imaging was of sufficient quality. No coronary aneurysms were detected in the acute phase or during follow-up.
We hope these clarifications improve the precision and transparency of our methodology.
Contradictory Statement on Statistical Methods
Reviewer 1 Comment: The statement:
"No advanced statistical methods were employed, as this study presents descriptive findings." appears unscientific, as even in descriptive studies, statistical creativity can be applied. Additionally, this statement contradicts the fact that the authors later mention using Fisher’s exact test and Mann-Whitney U test.
Author Response:
We thank the reviewer for this important observation. We agree that the original statement was inappropriate and potentially misleading. We have removed it and revised the relevant section to clearly describe the statistical methods used, including Fisher’s exact test and the Mann-Whitney U test, which were applied as appropriate to the data. We trust that this correction improves the rigor and clarity of the manuscript.
Misinterpretation of LVEF Data
Reviewer 1 Comment: Since the authors did not specify what LVEF value was considered normal/abnormal, the statement in lines 108–110:
- "with reduced left ventricular ejection fraction (LVEF) (median: 45%; IQR: 40–50%) detected in all seven cases." is concerning, as the IQR suggests that some patients had LVEF values that may be considered normal. Clarification is needed regarding what threshold was used for “reduced” LVEF, and whether the IQR refers to the entire sample.
- The issue can be addressed if the authors include LVEF in Table 2, renaming it to "laboratory and imaging findings" instead of "biomarker levels of..."
Author Response: We thank the reviewer for these helpful suggestions. We have clarified in the Methods section that we used the same LVEF thresholds as defined in the MUSIC study to determine normal versus reduced LVEF, ensuring consistency and comparability. In response to the concern about the reported IQR, we reanalyzed the data and updated the values accordingly to ensure they reflect only patients with reduced LVEF.
Given this clarification and the updated figures in the text, we believe it is no longer necessary to include the specific LVEF values in Table 2. Therefore, we have not added them to the table, but we are happy to do so should the editorial team prefer it. We also thank the reviewer for the suggestion to rename the table and have updated the title to “Laboratory and Imaging Findings” to better reflect its content.
Biomarker Correlation with Myocardial Dysfunction
Reviewer 1 Comment: Analyzing Table 2 and comparing it with data from lines 108–115, it appears that the authors report data from the entire sample (both with and without myocardial dysfunction), yet they write in lines 111–112:
"[those] levels [are] strongly correlated with myocardial involvement."
This is contradictory, as the data include patients without myocardial involvement. A comparison between groups would be more informative.
Author Response: We thank the reviewer for identifying this inconsistency. We agree that the statement was too general and could be misleading, as it did not clearly reflect the comparison between patient subgroups. We have removed the sentence to avoid confusion and improve the scientific accuracy of the manuscript.
Interpretations in the Results Section
Reviewer 1 Comment: In the Outcomes under Results, the authors state:
"Full recovery of myocardial function was observed in all patients, with no fatalities or residual cardiac dysfunction at follow-up. These findings underscore the efficacy of early and aggressive treatment strategies in mitigating severe outcomes."
This is problematic, as the study is non-interventional and does not support a causal relationship. It is recommended that the authors limit this section to objective findings.
Author Response: We fully agree with the reviewer’s comment. The statement indeed reflects an interpretation rather than a result, and our study design does not allow us to draw causal conclusions about the efficacy of treatment strategies. Therefore, we have removed this sentence from the Results section to maintain scientific rigor and avoid misinterpretation.
Number of Patients in the MUSIC Study in Table 3
Reviewer 1 Comment: In Table 3, how many patients were included in the MUSIC study?
Author Response: Thank you for pointing this out. We have added the number of patients included in the MUSIC study to Table 3 for clarity and completeness.
Figure 1 Requires Improved Graphical Representation
Reviewer 1 Comment: I recommend the authors remake Figure 1 using professional tools such as BioRender, BioArt, BioIcons, etc.
Author Response: We appreciate the reviewer’s suggestion. However, since Figure 1 was not essential to the manuscript and based on earlier comments, we have decided to remove it entirely to maintain focus and clarity.
Conclusions Not Supported by Data
Reviewer 1 Comment: In the Conclusion, the authors write: "Our study demonstrates that early and standardized interventions are highly effective in managing MIS-C, facilitating full recovery and preventing long-term complications."
This is misleading, as the study did not evaluate timing of intervention.
Author Response: We thank the reviewer for this important clarification. We agree that the original wording overstates the findings. The study design does not allow us to demonstrate causality regarding the timing or efficacy of interventions. We have therefore revised the sentence to replace “demonstrates” with “suggests,” making the conclusion more appropriately cautious and aligned with the study's observational nature.
Reviewer 1 Comment: The authors cannot confirm, based on their data, that:
"The absence of coronary artery aneurysms in our cohort supports the efficacy of aggressive early treatments—namely, intravenous immunoglobulin (IVIG) and corticosteroids."
The absence of aneurysms may be due to selection bias, small sample size, or chance. The study does not compare interventions and should not make such conclusions.
Author Response: We thank the reviewer for this valid observation. We agree that our study design does not allow us to attribute the absence of coronary aneurysms to the efficacy of any particular treatment strategy. We have revised the manuscript accordingly to clarify that this finding should be interpreted cautiously. The revised text now acknowledges potential limitations such as sample size and selection bias, and avoids drawing causal conclusions.
Reviewer 1 Comment: In the Conclusion, the statement: "Furthermore, vaccination emerged as a critical protective factor, significantly reducing both the incidence and severity of MIS-C."
is not supported by the study data and should be removed.
Author Response: We agree with the reviewer’s comment. As our study did not include vaccination data or comparisons between vaccinated and unvaccinated individuals, we have removed this statement from the Conclusion section.
Reviewer 1 – Minor Comment 1: In Table 2, clearly indicate—either in the table or in the legend—whether the data are presented as median (IQR) or another format.
Author Response: Thank you for this suggestion. We have clarified the format of the data in Table 2, specifying whether values are presented as percentages, medians, or interquartile ranges (IQR) as appropriate. Additionally, we corrected a typographical error where "men" was mistakenly written instead of "mean."
We sincerely appreciate your constructive feedback, which has significantly improved the quality and accuracy of our manuscript.
Best regards, The Authors

Reviewer 2 Report
Comments and Suggestions for Authors
The study is relevant since there is little information about Pediatric Inflammatory Multisystem Syndrome 3 (PIMS) Associated with SARS-CoV-2. The research is well conducted by the authors and the information is relevant to be applied in clinical practice. The authors evaluated myocardial dysfunction by echocardiogram and biomarkers of inflammation, cardiac function and coagulation (CRP, ferritin, NT-ProBNP, troponin and D-dimer). However, there are some issues in relation to the manuscript that should be improved:1. The title should be reconsidered since from my perspective it does not reflect the main objective of the study and is a more suitable title for a review article. I suggest the following title that in my opinion better reflects the objective of the work: "Effectiveness of early therapeutic interventions and vaccination strategies on myocardial dysfunction in patients with pediatric multisystem inflammatory syndrome associated with SARS-CoV-2." since myocardial dysfunction was assessed with echocardiogram and biomarkers.
2. The abstract should be restructured, for example, ferritin is not mentioned and it was a significant finding, it is mentioned that endothelial dysfunction was assessed; but in reality at least the most common biomarkers to assess endothelial dysfunction (nitrites, dimethyl-arginine, LDL-C, etc.) were not measured, so I suggest removing this part of the abstract.
3. Restructure the introduction, for example, it is not necessary to make a subdivision to indicate the molecular mechanisms and the information on the molecular mechanisms should be summarized in the introduction part.
4. In the results section, table 1 needs to be improved, since the information that is to be expressed is not understood, for example, it is not necessary to put the total number of patients in the study in the table since it is specified in the text, it is necessary to put subdivisions in the table in some sections, for example, in the gender section, make a division for men and women, in symptoms it is important to put the number of patients with each symptom, how many presented vomiting, diarrhea, etc.
It is not clear to me if IL-6 was evaluated, since it is mentioned in table 1, but in table 2 it is not mentioned, nor is its average value mentioned. Please clarify this situation. Was fibrinogen measured? It does not appear in Table 2. Please clarify this situation.What is the purpose of the table that is between lines 147-148? Since it is the same information as in table 1, review this information5. Review the discussion section and restructure. There is repeated information. For example, the information on lines 221-223, 234-237, 241-246 is repeated in other parts of the discussion with different words but the same idea. this situation needs to be reviewed.
5. The discussion should better address the aspect of biomarkers. Are there other studies that evaluate these biomarkers for this disease? If so, mention mean values, differences or similarities with what was observed in this study. Ferritin was a significant biomarker in Table 2 and is not addressed again in the discussion. I suggest adding it to the discussion, as this is interesting since ferritin has been associated with other situations such as ventilatory deterioration in patients with SARS-COV-2.
I think that figure 1 should be eliminated because it does not provide any new information. The information that this figure summarizes is well described in the text and is basically the same, which is why it seems redundant to me.
6. The conclusion should be precise and reflect the most important findings of the study. For example, the information in lines 331-337 should be the paragraph that closes the discussion section.
7. The English could be improved to more clearly express the research. For example, in table 2 PCR should be CRP, in table 1 diarrhea (instead of diarrhea), etc.
best regards
Comments on the Quality of English Language
The English could be improved to more clearly express the research. For example, in table 2 PCR should be CRP, in table 1 diarrhea (instead of diarrhea), etc.
Author Response
Response to Reviewer 2
Dear Reviewer,
We sincerely appreciate your insightful comments and the time you have dedicated to reviewing our manuscript. We have carefully considered your suggestions and have made the necessary revisions accordingly. Below, we provide responses to each of your comments and outline the corresponding modifications in the manuscript.
Reviewer 2 Comment: The English could be improved to more clearly express the research.
Author Response: We appreciate this observation and have reviewed the manuscript carefully to correct all detected language issues and improve clarity and readability throughout the text.
- Reviewer 2 Comment: The title should be reconsidered, as it does not reflect the main objective of the study and may be more suitable for a review article. Suggested title: “Effectiveness of early therapeutic interventions and vaccination strategies on myocardial dysfunction in patients with pediatric multisystem inflammatory syndrome associated with SARS-CoV-2.”
Author Response: We appreciate the reviewer’s suggestion. However, we respectfully disagree with the proposed title change. The second part of the suggested title, specifically the reference to vaccination strategies, may lead to confusion, especially considering the concerns raised by Reviewer 1 regarding the interpretation of vaccination data. As a result, we have already revised or removed such content to ensure the manuscript accurately reflects the scope and limitations of our study. Therefore, we believe retaining our original title—focused specifically on myocardial dysfunction in MIS-C patients—is more appropriate and aligned with the study’s objectives and data.
Reviewer 2 Comment: The abstract should be restructured. For example, ferritin is not mentioned even though it was a significant finding. Also, the reference to endothelial dysfunction should be removed, as specific biomarkers for it (e.g., nitrites, dimethyl-arginine, LDL-C) were not measured.
Author Response: We fully agree with the reviewer’s observations. We have removed the mention of endothelial dysfunction from the abstract, as our study did not include the appropriate biomarkers to assess it reliably. Additionally, we have included statistically significant findings from our series—such as ferritin—in the revised abstract. We also replaced the word “highlight” with “suggest” to better reflect the observational nature of our study and to avoid implying causal conclusions.
Reviewer 2 Comment: Restructure the introduction. It is not necessary to include a subdivision for molecular mechanisms, and that section should be summarized within the main introduction text.
Author Response: We appreciate the reviewer’s suggestion. We have restructured the Introduction by removing content related to molecular mechanisms that were not directly investigated in our study. Additionally, we revised the section to focus more clearly on the context and the most relevant aspects of our findings, ensuring that it aligns with the objectives and results of our research.
Reviewer 2 Comment: In the Results section, Table 1 needs to be improved. The total number of patients is already specified in the text, so it may not need to be repeated in the table. Subdivisions (e.g., gender: male/female; symptoms: number of patients with vomiting, diarrhea, etc.) should be clearly presented.
Also, it is unclear whether IL-6 and fibrinogen were evaluated. IL-6 appears in Table 1 but not in Table 2 or the text, and fibrinogen is not reported. Additionally, there is a table between lines 147–148 that appears to duplicate Table 1.
Finally, the Discussion section includes repetitive content that should be revised (e.g., lines 221–223, 234–237, and 241–246 repeat similar ideas).
Author Response: Thank you for these thoughtful and helpful suggestions.
- Table 1: We have revised Table 1 to improve clarity and structure. Subdivisions have been added (e.g., gender separated into male/female; symptoms listed individually with the number of patients affected). Although we initially believed that including the total number of patients helped contextualize the findings, we are open to removing it if deemed unnecessary by the editorial team.
- IL-6: IL-6 was measured in 4 out of the 15 patients, all of whom had cardiac dysfunction. In all 4 cases, IL-6 levels were elevated. This information has now been added to the appropriate table and clarified in the text.
- Fibrinogen: Fibrinogen was assessed in all patients. Fourteen out of fifteen showed elevated levels, but there were no significant differences between those with and without cardiac dysfunction (mean 801 mg/dL vs. 789 mg/dL). This has been added to Table 2 and described in the Results section.
- Duplicate Table: The table between lines 147–148 was mistakenly duplicated. We have removed the duplicate and improved Table 1 in accordance with all suggestions provided.ç
- Discussion Section: We have thoroughly reviewed and restructured the Discussion to eliminate redundancy and improve clarity. Repetitive content (including that found in lines 221–223, 234–237, and 241–246) has been consolidated or removed to ensure a more concise and focused narrative.
Reviewer 2 Comment: The Discussion should better address the role of biomarkers. Are there other studies that evaluated these biomarkers in MIS-C? Comparisons (e.g., mean values, similarities/differences) would be helpful. Also, although ferritin is a significant biomarker in Table 2, it is not mentioned in the discussion. This should be included, as ferritin has been associated with outcomes such as ventilatory deterioration in SARS-CoV-2 patients.
Additionally, Figure 1 appears redundant and could be removed.
Author Response: Thank you for these valuable suggestions.
We have added discussion of our ferritin findings in the Results, Conclusion, and Abstract sections. We agree that ferritin is a relevant biomarker in MIS-C and have included comparisons with findings from other studies that highlight its association with inflammation and disease severity in pediatric SARS-CoV-2 patients. If further elaboration in the Discussion is recommended, we would be happy to expand it further.
As for Figure 1, we agree that it was redundant given the content already described in the text. We have removed the figure from the manuscript as suggested.
Reviewer 2 Comment: The Conclusion should be precise and reflect the most important findings of the study. For example, the content in lines 331–337 would be more appropriate as a closing paragraph for the Discussion.
Author Response: We agree with the reviewer that the Conclusion section should better reflect the key findings of the study. We have revised it accordingly, summarizing the most relevant and statistically significant results. Additionally, we have ensured that the paragraph in lines 331–337 is now appropriately placed in the Discussion section to maintain logical structure and clarity.
Reviewer 2 Comment: The English could be improved to more clearly express the research. For example, in Table 2, “PCR” should be “CRP”; in Table 1, correct spelling of “diarrhea,” etc.
Author Response: Thank you for pointing this out. We have carefully reviewed the manuscript and corrected all identified typographical and language errors, including those mentioned in Tables 1 and 2, to ensure clarity and consistency throughout the text.
We sincerely appreciate your constructive feedback, which has significantly improved the quality and accuracy of our manuscript.
Best regards, The Authors

Reviewer 3 Report
Comments and Suggestions for Authors
I recommend the following:
- While the distinction between MIS-C and KD is well addressed later, briefly introducing this comparison in the introduction would help set the stage for readers unfamiliar with both conditions.
- Given the relatively small cohort (n=15), please provide a more detailed discussion on how this number impacts statistical power and generalizability.
- While the manuscript states that ethical approval and informed consent were obtained, please specify whether assent from older pediatric patients was sought, as per ethical standards.
- Some p-values (e.g., p > 0.05 in Table 2) indicate non-significant findings. It may be helpful to state how these results should be interpreted explicitly. Also, consider using confidence intervals to provide more insight into variability.
- While the manuscript notes differences between this study and the MUSIC cohort, could other factors (e.g., genetic predispositions, socioeconomic conditions) have influenced the absence of coronary aneurysms?
- Since follow-up data were collected for six months, did any patients experience lingering symptoms such as fatigue or cardiac dysfunction? If not assessed, please acknowledge this as a limitation.
- The study suggests that vaccination significantly reduces MIS-C severity. However, as only two vaccinated patients were included, a stronger disclaimer should clarify that this finding is consistent with broader literature rather than being a primary conclusion of this study.
- The study effectively highlights the need for standardized MIS-C treatment protocols. Please consider expanding on specific research directions, such as investigating potential genetic markers for MIS-C susceptibility or exploring cytokine-targeted therapies.
- Please include a visual comparison of biomarker levels between patients with and without myocardial dysfunction to enhance data interpretation.
Author Response
Reviewer 3 Comments and Author Responses
Dear Reviewer,
We sincerely appreciate your insightful comments and the time you have dedicated to reviewing our manuscript. We have carefully considered your suggestions and have made the necessary revisions accordingly. Below, we provide responses to each of your comments and outline the corresponding modifications in the manuscript.
Reviewer 3 Comment: 1. While the distinction between MIS-C and KD is well addressed later, briefly introducing this comparison in the introduction would help set the stage for readers unfamiliar with both conditions.
Author Response: Thank you for this suggestion. We have already included a brief comparative overview of cardiac involvement in MIS-C and Kawasaki disease in the Introduction to support reader understanding.
Reviewer 3 Comment: 2. Given the relatively small cohort (n=15), please provide a more detailed discussion on how this number impacts statistical power and generalizability.
Author Response: We agree and have expanded the Discussion and Conclusions to reflect how the small sample size limits statistical power and generalizability of our findings.
Reviewer 3 Comment: 3. While the manuscript states that ethical approval and informed consent were obtained, please specify whether assent from older pediatric patients was sought, as per ethical standards.
Author Response: We appreciate this observation. We have revised the Methods section to more clearly state that, in addition to informed consent, assent was requested from older pediatric participants when appropriate.
Reviewer 3 Comment: 4. Some p-values (e.g., p > 0.05 in Table 2) indicate non-significant findings. It may be helpful to state how these results should be interpreted explicitly. Also, consider using confidence intervals to provide more insight into variability.
Author Response: We have clarified the interpretation of non-significant findings in the Results section and have indicated which results reached statistical significance. Where applicable, we also incorporated confidence intervals to provide additional context for variability.
Reviewer 3 Comment: 5. While the manuscript notes differences between this study and the MUSIC cohort, could other factors (e.g., genetic predispositions, socioeconomic conditions) have influenced the absence of coronary aneurysms?
Author Response: Thank you for this thoughtful point. We have included in the Discussion a comment on the homogeneity of our sample—14 of the 15 patients were of Spanish origin—which may contribute to differences observed when compared with more diverse cohorts such as MUSIC. This has been noted as a potential explanatory factor for the absence of coronary aneurysms in our cohort.
Reviewer 3 Comment: 6. Since follow-up data were collected for six months, did any patients experience lingering symptoms such as fatigue or cardiac dysfunction? If not assessed, please acknowledge this as a limitation.
Author Response: We have addressed this in the Results and Limitations. All patients were followed up for six months, except for two who continued follow-up in other centers. Among those followed, no persistent fatigue or cardiac dysfunction was reported.
Reviewer 3 Comment: 7. The study suggests that vaccination significantly reduces MIS-C severity. However, only two vaccinated patients were included. A stronger disclaimer should clarify that this is based on broader literature.
Author Response: We agree with this observation. While we maintain a literature-based mention of vaccination in the Introduction for context, we have removed this conclusion from the Results, Discussion, and Conclusion sections, as our data do not support any vaccine-related claims.
Reviewer 3 Comment: 8. The study effectively highlights the need for standardized MIS-C treatment protocols. Please consider expanding on specific research directions.
Author Response: We appreciate this suggestion and have added to the Conclusion a brief discussion of future research directions, including exploring genetic markers for MIS-C susceptibility and evaluating cytokine-targeted therapies.
Reviewer 3 Comment: 9. Please include a visual comparison of biomarker levels between patients with and without myocardial dysfunction.
Author Response: Thank you. We believe that this comparison is adequately reflected in Table 2, which includes key biomarker data, now expanded to include fibrinogen following another reviewer’s suggestion.
We sincerely appreciate your constructive feedback, which has significantly improved the quality and accuracy of our manuscript.
Best regards, The Authors

Round 2
Reviewer 1 Report
Comments and Suggestions for Authors
I thank the authors for their thorough review. I believe the manuscript now conveys the authors' intended message much more clearly.
Minor suggestions:
-
I recommend shortening the "Conclusion" section to a maximum of two paragraphs: one summarizing the results, and the other highlighting the discussion and key take-home messages.
Author Response
Response to Reviewer 1 (Second Round)
Comment: I recommend shortening the “Conclusion” section to a maximum of two paragraphs: one summarizing the results, and the other highlighting the discussion and key take-home messages.
Response: We appreciate your suggestion and have revised the Conclusion accordingly. It now includes two concise paragraphs: the first summarizes the study’s main findings, and the second presents the key conclusions and broader implications.
Conclusions
In our series, managed uniformly according to international guidelines, nearly half of the patients presented with ventricular dysfunction that fully resolved without mortality or coronary artery complications. Parameters of inflammation, myocardial damage, and coagulopathy were universally elevated, with significant differences in ferritin, NT-proBNP, and troponin levels between patients with and without ventricular dysfunction.
These findings emphasize the effectiveness of early and standardized interventions—namely intravenous immunoglobulin and corticosteroids—in achieving complete recovery and preventing long-term cardiac complications in MIS-C. The absence of coronary aneurysms in our cohort supports the value of timely treatment, although the limited sample size should be acknowledged. Future studies should focus on biomarker-guided management, long-term outcomes, and the potential role of genetic susceptibility and cytokine-targeted therapies in refining MIS-C treatment strategies.

Reviewer 2 Report
Comments and Suggestions for Authors
The article has improved substantially; however, I have some observations.
1. The introduction should not have subdivisions; it should be a continuous text.
2. The discussion should focus entirely on the role of biomarkers in the disease (NT-ProBN, troponin, and ferritin), especially since the title of the work focuses on molecular mechanisms, and the discussion section only covers lines 298-311. Therefore, I believe this section should be expanded.
Best regards.
Author Response
Response to Reviewer 2 (Second Round)
Comment 1: The introduction should not have subdivisions; it should be a continuous text.
Response: Thank you for your valuable feedback. We have revised the Introduction to remove the “Molecular Pathophysiology” subsection. The entire section is now presented as a unified and continuous narrative to improve clarity and cohesion.
Comment 2: The discussion should focus more on the role of biomarkers (NT-ProBNP, troponin, and ferritin), particularly given the molecular emphasis of the manuscript title. The current discussion covers only lines 298–311.
Response: We completely agree. We have substantially expanded the section of the Discussion dedicated to biomarkers, particularly NT-ProBNP, troponin, and ferritin. We have also included references to other studies in which these markers are elevated in severe COVID-19 and MIS-C, as well as additional observations regarding thrombocytopenia, even though the difference was not statistically significant. This revision strengthens the alignment between our findings and the molecular focus of the manuscript.
“The elevated levels of NT-proBNP and troponin observed in patients with myocardial dysfunction in our cohort indicate acute myocardial stress and injury, which are hallmark features of MIS-C. These biomarkers serve as critical tools for early recognition of cardiac involvement and reflect the transient, cytokine-driven myocardial depression that characterizes this syndrome [Valverde et al, 2021; Dionne et al, 2020]. In contrast to Kawasaki disease, where myocardial damage is often linked to persistent coronary inflammation and vascular remodeling, MIS-C typically presents with rapid onset of dysfunction and recovery following immunomodulatory therapy [McCrindle & Manlhiot 2020]. Ferritin levels were also elevated among patients with myocardial involvement, supporting its role as an inflammatory marker in MIS-C. Elevated ferritin has been associated with macrophage activation syndrome and worse outcomes in both severe COVID-19 and MIS-C [Eloseily et al., 2020; Abrams et al., 2021]. Its increase may indicate systemic hyperinflammation and correlate with disease severity. Although the difference in platelet counts between groups was not statistically significant, a trend toward thrombocytopenia was observed in the myocardial dysfunction group, aligning with findings from other MIS-C studies describing platelet consumption in inflammatory states [Belhadjer et al., 2020]. These biomarker profiles underscore the immunopathogenic differences between MIS-C and Kawasaki disease. While KD involves chronic endothelial activation and a significant risk of coronary aneurysms, MIS-C is characterized by acute elevations in proinflammatory cytokines—such as IL-6, IL-1β, and TNF-α—leading to systemic inflammation and transient cardiac dysfunction [Lee et al., 2020; Rowley 202]. The absence of coronary aneurysms in our cohort despite evident myocardial involvement supports the hypothesis of a rapid, cytokine-mediated inflammatory response rather than sustained vascular injury.”
New 8 references to be included in this section
Valverde I, Singh Y, Sanchez-de-Toledo J, Theocharis P, Chikermane A, Di Filippo S, Kuciñska B, Mannarino S, Tamariz-Martel A, Gutierrez-Larraya F, Soda G, Vandekerckhove K, Gonzalez-Barlatay F, McMahon CJ, Marcora S, Napoleone CP, Duong P, Tuo G, Deri A, Nepali G, Ilina M, Ciliberti P, Miller O; AEPC COVID-19 Rapid Response Team*. Acute Cardiovascular Manifestations in 286 Children With Multisystem Inflammatory Syndrome Associated With COVID-19 Infection in Europe. Circulation. 2021 Jan 5;143(1):21-32. doi: 10.1161/CIRCULATIONAHA.120.050065. Epub 2020 Nov 9. PMID: 33166189.
Dionne A, Mah DY, Son MBF, Lee PY, Henderson L, Baker AL, de Ferranti SD, Fulton DR, Newburger JW, Friedman KG. Atrioventricular Block in Children With Multisystem Inflammatory Syndrome. Pediatrics. 2020 Nov;146(5):e2020009704. doi: 10.1542/peds.2020-009704. Epub 2020 Aug 27. PMID: 32855347.
McCrindle BW, Manlhiot C. SARS-CoV-2-Related Inflammatory Multisystem Syndrome in Children: Different or Shared Etiology and Pathophysiology as Kawasaki Disease? JAMA. 2020 Jul 21;324(3):246-248. doi: 10.1001/jama.2020.10370. PMID: 32511667.
Eloseily EM, Weiser P, Crayne CB, Haines H, Mannion ML, Stoll ML, Beukelman T, Atkinson TP, Cron RQ. Benefit of Anakinra in Treating Pediatric Secondary Hemophagocytic Lymphohistiocytosis. Arthritis Rheumatol. 2020 Feb;72(2):326-334. doi: 10.1002/art.41103. Epub 2019 Dec 26. PMID: 31513353.
Abrams JY, Oster ME, Godfred-Cato SE, Bryant B, Datta SD, Campbell AP, Leung JW, Tsang CA, Pierce TJ, Kennedy JL, Hammett TA, Belay ED. Factors linked to severe outcomes in multisystem inflammatory syndrome in children (MIS-C) in the USA: a retrospective surveillance study. Lancet Child Adolesc Health. 2021 May;5(5):323-331. doi: 10.1016/S2352-4642(21)00050-X. Epub 2021 Mar 10. PMID: 33711293; PMCID: PMC7943393.
Belhadjer Z, Méot M, Bajolle F, Khraiche D, Legendre A, Abakka S, Auriau J, Grimaud M, Oualha M, Beghetti M, Wacker J, Ovaert C, Hascoet S, Selegny M, Malekzadeh-Milani S, Maltret A, Bosser G, Giroux N, Bonnemains L, Bordet J, Di Filippo S, Mauran P, Falcon-Eicher S, Thambo JB, Lefort B, Moceri P, Houyel L, Renolleau S, Bonnet D. Acute Heart Failure in Multisystem Inflammatory Syndrome in Children in the Context of Global SARS-CoV-2 Pandemic. Circulation. 2020 Aug 4;142(5):429-436. doi: 10.1161/CIRCULATIONAHA.120.048360. Epub 2020 May 17. PMID: 32418446.
Lee PY, Day-Lewis M, Henderson LA, Friedman KG, Lo J, Roberts JE, Lo MS, Platt CD, Chou J, Hoyt KJ, Baker AL, Banzon TM, Chang MH, Cohen E, de Ferranti SD, Dionne A, Habiballah S, Halyabar O, Hausmann JS, Hazen MM, Janssen E, Meidan E, Nelson RW, Nguyen AA, Sundel RP, Dedeoglu F, Nigrovic PA, Newburger JW, Son MBF. Distinct clinical and immunological features of SARS-CoV-2-induced multisystem inflammatory syndrome in children. J Clin Invest. 2020 Nov 2;130(11):5942-5950. doi: 10.1172/JCI141113. PMID: 32701511; PMCID: PMC7598077.
